# Tailoring automated data augmentation to H&E-stained histopathology

**Khrystyna Faryna**[1]                                 KHRYSTYNA.FARYNA@RADBOUDUMC.NL
**Jeroen van der Laak**[1]                          JEROEN.VANDERLAAK@RADBOUDUMC.NL
**Geert Litjens**[1]                                        GEERT.LITJENS@RADBOUDUMC.NL
[1] *Department of Pathology, Radboud University Medical Center, Nijmegen, the Netherlands*

## Abstract

Convolutional neural networks (CNN) are sensitive to domain shifts, which can result in poor generalization. In medical imaging, data acquisition conditions differ among institutions, which leads to variations in image properties and thus domain shift. Stain variation in histopathological slides is a prominent example. Data augmentation is one way to make CNNs robust to varying forms of domain shift but requires extensive hyper-parameter tuning. Due to the large search space, this is cumbersome and often leads to sub-optimal generalization performance. In this work, we focus on automated and computationally efficient data augmentation policy selection for histopathological slides. Building upon the RandAugment framework, we introduce several domain-specific modifications relevant to histopathological images, increasing generalizability. We test these modifications on H&E-stained histopathology slides from Camelyon17 dataset. Our proposed framework outperforms the state-of-the-art manually engineered data augmentation strategy, achieving an area under the ROC curve of 0.964 compared to 0.958, respectively.

**Keywords:** computational pathology, data augmentation, autoML, domain shift

## 1. Introduction

A key issue with vanilla deep neural networks is poor generalizability in the face of domain shift. In medical imaging, domain shift can occur when testing on data from different centers, where each center has its own acquisition protocols. This is especially prominent in computational pathology due to multiple sources of shift. In the process of obtaining whole-slide images (WSI) the tissue is stained with various dyes, the most common being hematoxylin and eosin (H&E). While the staining process enables the underlying structure of tissues to be visible under the microscope, at the same time it is a source of variation in image contrast and color due to differences in chemicals and staining protocols. Secondly, after staining, slides are digitized with a scanner, which, depending on the brand, has different optical characteristics and post-processing filters. Figure 1 illustrates variation among examples of WSI patches originating from different institutions. The image samples were taken from Camelyon17 challenge dataset (Bándi et al., 2019) for detection of tumor metastasis in breast lymph nodes.

The image properties variation among different centers can heavily degrade the performance of machine learning algorithms that were trained on the data from a single institution when testing on slides from external centers. Developing frameworks robust to changes in image acquisition parameters is essential to achieve generalization.

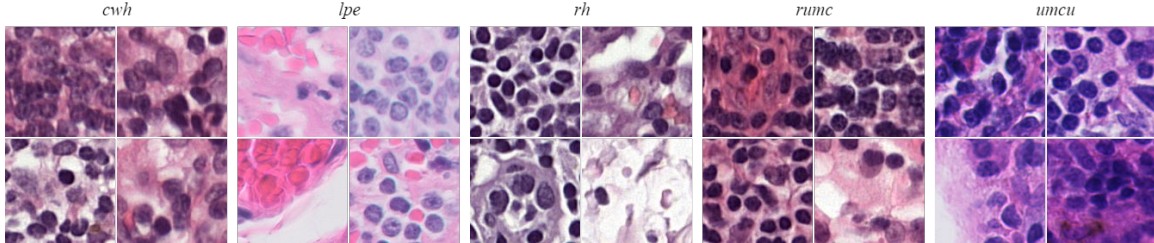

Figure 1: Stain variation among different centers in tumour metastasis detection in breast lymph node tissue resections. Examples of WSI patches originating from different institutions (from left to write): *cwh, lpe, rh, rumc, umcu.*

The common solutions to increase the robustness of machine learning models could be divided into three categories: mapping the data into domain invariant space, stain normalization, and data augmentation.

Lafarge et al. (2019); Graziani et al. (2020) utilized domain-adversarial training to learn domain-invariant features for H&E slides. The authors involve an additional classifier in the training pipeline (domain discriminator) and jointly optimize the weights of both, so that the domain-specific features, which are not beneficial for the main task, would be removed from the representation. A disadvantage is that it requires modification of the training architecture, and it is not always possible to ensure robustness to variation which is not in the data used for domain-adversarial training.

Stain normalization is a group of techniques that aim to match the color properties of the test data to those of training data. Stain normalization methods can be subdivided into two categories: classical and machine learning-based. The classical ones, often based on physics (Macenko et al., 2009), rely on statistical analysis to match the properties of images from source and target domains (Reinhard et al., 2001). The machine learning-based methods, like (Ehteshami Bejnordi et al., 2016), focus on classifying pixels into different stain components, subsequently matching the characteristic distributions of source and target slides. Alternatively, the stain normalization could be viewed as a style transfer problem, where the data from an external institution is transformed to the domain of the target center (de Bel et al., 2019; Swiderska-Chadaj et al., 2020). An advantage of normalization is that it decouples the tasks of domain invariance and classification, but a disadvantage is that an extra step is required to normalize each image before it can be classified.

Data augmentation is a frequently used solution to introduce a certain degree of invariance to a CNN (Shorten and Khoshgoftaar, 2019; Saha et al., 2020b), it also can be used to tackle class imbalance (Sebai et al., 2020). Augmentation can regularize training by introducing distortion to data and preventing overfitting, moreover, it also has the capacity to introduce prior knowledge to the model (Goyal and Bengio, 2020; Saha et al., 2020a). Nevertheless, manual selection of augmentation parameters is cumbersome: extremely large number of augmentation hyper-parameters and their combinations lead to optimal performance often being overlooked during the hyper-parameter tuning process.

Recently a number of automatic data augmentation policy selection methods have been proposed. AutoAugment (Cubuk et al., 2019) is a framework for automated augmentation policy selection. Based on reinforcement learning, it relies on a controller RNN to predict an augmentation policy from the predefined search space. A daughter network with a fixed set of parameters and selected augmentation policy is trained until convergence achieving accuracy R. This value R is then used as a reward to update the gradients of the controller, thus enabling it to generate more beneficial policies as training progresses. Despite achieving state-of-art performance on several public benchmarks, due to the extreme compute cost, its application to other tasks is challenging.

Fast AutoAugment (Lim et al., 2019), which treats augmented images as missing data points, has been proposed as an alternative, reducing computational time. These missing data points are recovered through a set of inference-time augmentations and Bayesian optimization.

RandAugment (Cubuk et al., 2020) focuses on reducing the search space and subsequently eliminating the requirement for a separate proxy task. The authors define a set of $K$ transforms, each transforms has a uniform probability to be selected ($\frac{1}{K}$). The final search space of the method consists of a linearly parameterized set of magnitudes $m$ and number of sequentially applied transforms $n$. The authors subsequently propose to find optimal parameters $m_{opt.}$ and $n_{opt.}$ for a particular dataset and network through grid-search.

In this study, we aim to design a framework for *fast* and *automatic* selection of an optimal data augmentation strategy suitable for computational pathology, namely H&E stained WSI. We compare against the current state-of-the-art for data augmentation in computational pathology: the extensive review of various augmentation strategies by Tellez et al. (2019). The authors analyze the performance impact of both classical and domain-specific augmentations and show that data augmentation alone (without stain normalization) can achieve superior performance on multi-institutional datasets.

The contributions of this work are following:

- We perform several modifications to RandAugment, enabling models to generalize over data originating from external institutions in computational pathology. In particular, we focus on increasing the robustness of a model to variations in color properties of the images resulting from different H&E staining protocols and scanners (Leo et al., 2016).

- We introduce an automated data augmentation framework that allows quickly finding optimal data augmentation parameters for a particular problem in computational pathology, in particular for H&E stained WSI.

- Our proposed framework outperforms current state-of-the-art manually engineered augmentation strategy on average.

- The proposed framework can potentially be scaled and adjusted to other tasks and datasets in H&E stained histopathology via tuning only two hyperparameters.

Code is available at https://github.com/DIAGNijmegen/pathology-he-auto-augment.

## 2. Methodology

### 2.1. Data

In this work, we focus on tumor metastasis detection in breast lymph node resections. We use the data from publicly available Camelyon17 challenge (Bándi et al., 2019). The dataset consists of 50 H&E stained WSI with the resolution of 0.25 $\mu$m/pixel, where experts annotated the metastases. The WSI originate from 5 different institutions, here denoted as *rumc*, *umcu*, *cwh*, *rh*, *lpe*. A total of approximately 1.5M patches have been extracted from the WSI. We formulate the cancerous cell detection task in WSI as a binary classification of WSI patches.

The original RandAugment framework relies on the assumption that training, validation, and test data come from the same distribution. Thus, improving scores on validation data means achieving a better generalization. On the other hand, if we aim to achieve a cross-dataset generalization, using validation scores coming from a single institution is unreliable due to overfitting.

In this work, we assume that a model trained on data from institution $a$ validated on centers $b, c$ is capable of generalizing to data from other unseen institutions. Thus, we arrange the experiments in the following way: we always train the model only on data from *rumc*, the validation set contains a subset of *rumc* and two external institutions, while the test set consists of data from the remaining two centers. The datasets in validation and testing are subsequently permuted resulting in six possible unique combinations.

### 2.2. H&E tailored RandAugment

Building upon RandAugment (Cubuk et al., 2020), we exploit the capacity of data augmentation to encode prior knowledge about presumable natural variation of images, thereby introducing a certain degree of invariance to the network. Stain variation among different centers is often represented as color feature changes in WSI. Firstly, we append the list of transforms with two histopathology-specific augmentations: random shifts in hematoxylin-eosin-DAB (*HED*) (Tellez et al., 2018) and hue-saturation-value (*HSV*) color spaces. Secondly, we excluded the *'posterize'*, *'solarize'* and *'invert'* as those result in unrealistic image appearances.

The full set of transforms used in this study along with their corresponding sets of magnitude ranges is shown in Table 1, the visual examples are shown in Figure 2.

Table 1: A set of augmentation transforms used in this study along with their corresponding ranges.

| transform type | magnitude range | transform type | magnitude range |
|---|---|---|---|
| *'identity'* | - | *'shear x'* | [-0.9, 0.9] |
| *'contrast'* | [0.0, 5.5] | *'shear y'* | [-0.9, 0.9] |
| *'brightness'* | [0.0, 5.5] | *'HED shift'* | [-0.9, 0.9] |
| *'sharpness'* | [0.0, 5.5] | *'HSV shift'* | [-0.9, 0.9] |
| *'rotation'* | [-90.0, 90.0] | *'autocontrast'* | - |
| *'translate x'* | [-30.0, 30.0] | *'color'* | [0.0, 5.5] |
| *'translate y'* | [-30.0, 30.0] | *'equalize'* | - |

Same as in RandAugment, for each of the parametrized transforms we define a linear discretization, $m$, between minimum and maximum values of the range. In this study, we explore the ranges: m = {0:15} and n = {0:3}, where n is a number of sequentially applied transforms per sample.

Unlike the original implementation of RandAugment we found that setting a global optimal *constant* value, $m_{opt.}$, for magnitudes of all the transforms to be insufficient in our particular task.

Thus, we redefine the hyper-parameter, $m$, such that every time a transform is applied, its magnitude, $M$ is selected as follows:

$$M \sim U(0, m_{opt.}), \quad m_{opt.} = \max_{\{m,n\}}(AUC_{val}),$$

where, $m$ is a set of upper bounds of the magnitudes, $n$ is a set of numbers of sequentially applied transforms. The $n_{opt.}$ and $m_{opt.}$, optimal values of $m$ and $n$ correspondingly, are found through grid-search as proposed in the RandAugment paper, with the monitoring quantity being the validation AUC. We found $n_{opt.} = 3$ and $m_{opt.} = 5$ to be suitable for our setup.

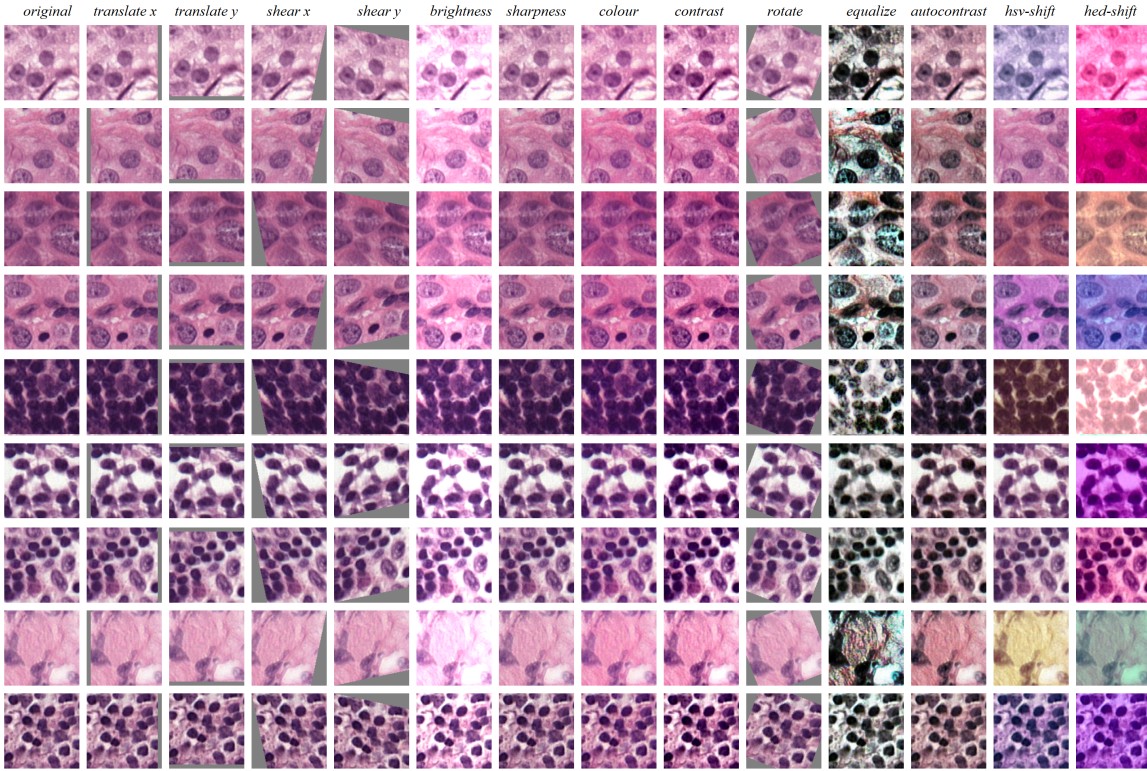

Figure 2: Example of augmentations with $m = 7$.

## 2.3. Experimental setup

As a baseline for comparison, we adopt the highest performing augmentation method analyzed in Tellez et al. (2019), namely 'hed-light': a manually tuned combination of classical and domain-specific data augmentation methods. Additionally, we also compare to not using any augmentation at all. In this study, we used the same architecture (Table 2) and training setup as proposed by Tellez et al. (2019) to make the comparison as fair as possible.

For all the experiments we used the cross-entropy loss function, Adam optimizer, class-balanced batches of 64 patches, patch size of 128x128 px., and a learning rate of $1 * 10^{-2}$, decaying by a factor of 10 based on the validation loss with a patience of 4 epochs. We apply $1 * 10^{-6}$ factor $L_2$ regularization as in Tellez et al. (2019). The training was stopped after the learning rate dropped to $1 * 10^{-5}$, the model with the lowest validation loss was selected as final.

Table 2: CNN architecture

| layer | # filters | stride | kernel size | batch norm | activation | padding | output |
|---|---|---|---|---|---|---|---|
| Conv2D | 32 | 1 | 3x3 | true | leaky-ReLU | valid | (128,128,32) |
| Conv2D | 64 | 2 | 3x3 | true | leaky-ReLU | valid | (126,126,64) |
| Conv2D | 64 | 1 | 3x3 | true | leaky-ReLU | valid | (62,62,64) |
| Conv2D | 128 | 2 | 3x3 | true | leaky-ReLU | valid | (60,60,128) |
| Conv2D | 128 | 1 | 3x3 | true | leaky-ReLU | valid | (29,29,128) |
| Conv2D | 256 | 2 | 3x3 | true | leaky-ReLU | valid | (27,27,256) |
| Conv2D | 256 | 1 | 3x3 | true | leaky-ReLU | valid | (13,13,256) |
| Conv2D | 512 | 2 | 3x3 | true | leaky-ReLU | valid | (11,11,512) |
| Conv2D | 512 | 1 | 3x3 | true | leaky-ReLU | valid | (5,5,512) |
| Conv2D | 512 | 2 | 3x3 | true | leaky-ReLU | valid | (3,3,512) |
| GlobalAveragePooling2D | | | | | | | |
| Dropout(0.5) | | | | | | | |
| Dense | 512 | | | true | leaky-ReLU | | (1,512) |
| Dense | 2 | | | false | softmax | | (1,2) |

## 3. Results

The performance of classification models was evaluated using the area under receiver operating curve (AUC). Bootstrapping with 10,000 iterations was used to compute p-values; statistical significance was assumed at $p < 0.05$.

Models trained on a dataset from a single institution with no data augmentation show generally poor performance on data from external centers. Table 3 in detail shows AUC scores from all of the folds of the model trained with no data augmentation.

The classification performance of the model trained with baseline (Tellez et al., 2019) augmentation is shown on the Table 4, while Table 5 shows the performance of the model trained with H&E tailored RandAugment.

Table 3: Classification performance obtained with no data augmentation, subscript indicates standard deviation.

| data split | | validation AUC | | | | test AUC | | | |
|---|---|---|---|---|---|---|---|---|---|
| val. set | test set | cwh | lpe | rh | umcu | cwh | lpe | rh | umcu |
| cwh+lpe | rh+umcu | 0.893 | 0.285 | - | - | - | - | 0.836 | 0.110 |
| cwh+rh | lpe+umcu | 0.903 | - | 0.743 | - | - | 0.221 | - | 0.104 |
| rh+lpe | cwh+umcu | - | 0.202 | 0.658 | - | 0.801 | - | - | 0.115 |
| umcu+cwh | lpe+rh | 0.751 | - | - | 0.106 | - | 0.262 | 0.640 | - |
| umcu+lpe | rh+cwh | - | 0.690 | - | 0.838 | 0.682 | - | 0.616 | - |
| umcu+rh | lpe+cwh | - | - | 0.640 | 0.115 | 0.736 | 0.282 | - | - |
| **average AUC/set:** | | $0.849_{0.085}$ | $0.392_{0.261}$ | $0.680_{0.055}$ | $0.352_{0.420}$ | $0.740_{0.060}$ | $0.255_{0.031}$ | $0.697_{0.121}$ | $0.108_{0.006}$ |

The average performance on external test sets of models trained with no augmentation, manually tuned augmentation proposed by (Tellez et al., 2019) and H&E tailored RandAugment is shown on Table 6. Additionally, a comparison with RandAugment using a constant value of m (as in Cubuk et al. (2020)) and stain normalization is provided in Appendix A and B correspondingly.

Table 4: Classification performance obtained with the highest performing augmentation strategy (*hed-light*) proposed by (Tellez et al., 2019), subscript indicates standard deviation.

| data split | | validation AUC | | | | test AUC | | | |
|---|---|---|---|---|---|---|---|---|---|
| val. set | test set | *cwh* | *lpe* | *rh* | *umcu* | *cwh* | *lpe* | *rh* | *umcu* |
| *cwh+lpe* | *rh+umcu* | 0.961 | 0.975 | - | - | - | - | 0.945 | 0.970 |
| *cwh+rh* | *lpe+umcu* | 0.959 | - | 0.955 | - | - | 0.975 | - | 0.960 |
| *rh+lpe* | *cwh+umcu* | - | 0.973 | 0.940 | - | 0.946 | - | - | 0.965 |
| *umcu+cwh* | *lpe+rh* | 0.952 | - | - | 0.974 | - | 0.973 | 0.944 | - |
| *umcu+lpe* | *rh+cwh* | - | 0.975 | - | 0.970 | 0.955 | - | 0.941 | - |
| *umcu+rh* | *lpe+cwh* | - | - | 0.948 | 0.977 | 0.959 | 0.963 | - | - |
| **average AUC/set:** | | $0.957_{0.005}$ | $0.975_{0.001}$ | $0.948_{0.008}$ | $0.974_{0.004}$ | $0.953_{0.007}$ | $0.970_{0.006}$ | $0.943_{0.002}$ | $0.965_{0.005}$ |

Table 5: Classification performance obtained with H&E-tailored RandAugment, subscript indicates standard deviation.

| data split | | validation AUC | | | | test AUC | | | |
|---|---|---|---|---|---|---|---|---|---|
| val. set | test set | *cwh* | *lpe* | *rh* | *umcu* | *cwh* | *lpe* | *rh* | *umcu* |
| *cwh+lpe* | *rh+umcu* | 0.969 | 0.957 | - | - | - | - | 0.952 | 0.984 |
| *cwh+rh* | *lpe+umcu* | 0.965 | - | 0.948 | - | - | 0.957 | - | 0.978 |
| *rh+lpe* | *cwh+umcu* | - | 0.971 | 0.957 | - | 0.965 | - | - | 0.984 |
| *umcu+cwh* | *lpe+rh* | 0.967 | - | - | 0.981 | - | 0.949 | 0.949 | - |
| *umcu+lpe* | *rh+cwh* | - | 0.962 | - | 0.983 | 0.967 | - | 0.953 | - |
| *umcu+rh* | *lpe+cwh* | - | - | 0.948 | 0.978 | 0.968 | 0.957 | - | - |
| **average AUC/set:** | | $0.967_{0.002}$ | $0.964_{0.007}$ | $0.951_{0.005}$ | $0.981_{0.002}$ | $0.967_{0.002}$ | $0.954_{0.005}$ | $0.951_{0.002}$ | $0.982_{0.003}$ |

Table 6: Average classification performance on external test sets of models trained with different augmentation settings, subscript indicates standard deviation.

| | no augmentation | baseline (Tellez et al., 2019) | H&E tailored RandAugment |
|---|---|---|---|
| **average test AUC:** | $0.450_{0.316}$ | $0.958_{0.013}$ | $0.964_{0.014}$ |

Overall, models trained with both augmentation methods achieve a good degree of generalization over data coming from external institutions. A model trained with H&E tailored RandAugment significantly outperforms ($p < 0.05$) the manually tuned framework, achieving an average 0.964 test AUC over 0.958 test AUC correspondingly. The H&E tailored RandAugment outperforms the baseline on 3 out of 4 external test sets: *cwh, rh* and *umcu*. On the other hand, the manually tuned framework archives a higher score on *lpe* test set.

## 4. Discussion

Automated frameworks offer a more structured and methodological approach to data augmentation. In medical imaging, these frameworks could be particularly beneficial if combined with domain-specific knowledge. Supplementing a set of transforms with augmentations in HSV and HED spaces is essential to improve generalization of a CNN in computational pathology.

The proposed H&E-tailored RandAugment outperforms the manually tuned baseline on 3 out of 4 external test sets, while the manually tuned baseline achieved a higher AUC score on *lpe* test set. We assume that this could be caused by the additional augmentations used in the baseline, e.g. Gaussian noise and elastic deformation, which might be beneficial for histopathology.

The RandAugment method relies on grid-search across a set of magnitudes and sequentially applied transforms to find the optimal augmentation strategy for a particular dataset and network. The results presented in RandAugment suggest that using a single *constant* value of a magnitude hyper-parameter, $m$, (found through grid-search) across all of the transforms in the set is sufficient to obtain the state-of-art performance on general computer vision benchmarks. Our experiments suggest that using a random value magnitude bounded between zero and $m_{opt.}$, where $m_{opt.}$ is the optimal upper bound found through grid-search, can improve classification performance in computational pathology. We assume that such behavior could be simply related to the fact that histopathology datasets often have a comparatively small number of independent samples and setting a range of values rather than a constant value for a transform magnitude provides a better regularization for a CNN.

In future work we aim to include additional augmentations in the RandAugment framework and compare against other automated data augmentation strategies, such as fast-autoaugment etc. Additionally, we want to benchmark across more multi-center digital pathology datasets.

Summarizing, in this work we present a method for a fast automatic selection of optimal data augmentation for H&E stained histopathology slides, outperforming the current state-of-the-art.

## Acknowledgments

The authors would like to thank David Tellez for his help with setting up the framework and baseline augmentation. The collaboration project is co-funded by the PPP Allowance made available by Health Holland, Top Sector Life Sciences & Health, to stimulate public-private partnerships.

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

## Appendix A. Constant magnitude in RandAugment

In original implementation of RandAugment (Cubuk et al., 2020) a constant value of m (found through grid-search) is used. In this experiment we use a *constant* value of m, as proposed in (Cubuk et al., 2020). The classification results are shown on Table 7, the model achieves an average AUC of $0.942_{0.028}$ on test sets.

Table 7: Classification performance obtained with constant value of $m_{opt} = 3$ of H&E tailored RandAugment augmentation strategy, subscript indicates standard deviation.

| data split | | validation AUC | | | | test AUC | | | |
|---|---|---|---|---|---|---|---|---|---|
| val. set | test set | *cwh* | *lpe* | *rh* | *umcu* | *cwh* | *lpe* | *rh* | *umcu* |
| *cwh+lpe* | *rh+umcu* | 0.965 | 0.942 | - | - | - | - | 0.931 | 0.913 |
| *cwh+rh* | *lpe+umcu* | 0.967 | - | 0.949 | - | - | 0.898 | - | 0.968 |
| *rh+lpe* | *cwh+umcu* | - | 0.961 | 0.958 | - | 0.960 | - | - | 0.977 |
| *umcu+cwh* | *lpe+rh* | 0.962 | - | - | 0.978 | - | 0.923 | 0.952 | - |
| *umcu+lpe* | *rh+cwh* | - | 0.920 | - | 0.982 | 0.963 | - | 0.945 | - |
| *umcu+rh* | *lpe+cwh* | - | - | 0.920 | 0.982 | 0.966 | 0.885 | - | - |
| **average AUC/set:** | | $0.965_{0.003}$ | $0.941_{0.0211}$ | $0.942_{0.020}$ | $0.981_{0.002}$ | $0.963_{0.003}$ | $0.902_{0.019}$ | $0.943_{0.011}$ | $0.959_{0.040}$ |

## Appendix B. Stain normalization

We additionally include results of a model trained with no data augmentation, using only stain normalization, while the rest of the training setup and parameters remain unchanged. In this experiment we use stain normalization method proposed by Tellez et al. (2019): a technique based on U-Net that translates augmented versions of the image to a target normalized one. The classification results are shown on Table 8, the model achieves an average AUC of $0.893_{0.067}$ on test sets.

Table 8: Classification performance obtained with no data augmentation and normalization method proposed by (Tellez et al., 2019), subscript indicates standard deviation.

| data split | | validation AUC | | | | test AUC | | | |
|---|---|---|---|---|---|---|---|---|---|
| val. set | test set | cwh | lpe | rh | umcu | cwh | lpe | rh | umcu |
| cwh+lpe | rh+umcu | 0.921 | 0.942 | - | - | - | - | 0.878 | 0.968 |
| cwh+rh | lpe+umcu | 0.918 | - | 0.865 | - | - | 0.918 | - | 0.959 |
| rh+lpe | cwh+umcu | - | 0.937 | 0.796 | - | 0.876 | - | - | 0.969 |
| umcu+cwh | lpe+rh | 0.847 | - | - | 0.967 | - | 0.939 | 0.780 | - |
| umcu+lpe | rh+cwh | - | 0.934 | - | 0.968 | 0.863 | - | 0.786 | - |
| umcu+rh | lpe+cwh | - | - | 0.790 | 0.963 | 0.850 | 0.928 | - | - |
| **average AUC/set:** | | $0.895_{0.042}$ | $0.938_{0.004}$ | $0.817_{0.042}$ | $0.966_{0.003}$ | $0.863_{0.013}$ | $0.928_{0.011}$ | $0.815_{0.054}$ | $0.965_{0.006}$ |

