# OpenReview forum: "Tailoring automated data augmentation to H&E-stained histopathology"
_MIDL.io/2021/Conference — MIDL 2021_

### Official Review · AnonReviewer3 · 2021-03-02

**Confidence:** 5
**Preliminary Rating:** 3
**Recommendation:** Oral, Poster
**Final Rating:** 4

**Summary:**

The paper describes a method for automatic data augmentation for H&E stained images based on the RandAugment framework. The proposed method was applied for an image classification task based on the Camelyon 2017 dataset. The result section shows the superior performance of the proposed method in comparison to a state-of-the-art hand-crafted augmentation technique.

**Strengths:**

- The paper is well-written as easy to follow in most parts.
- The research topic is very interesting and the proposed method can be adapted for various image processing tasks in H&E-stained images.
- The use of the proposed modified RandAugment framework for H&E-stained images is a novel approach.

**Weaknesses:**

There are no major weaknesses in the paper. However, some parts are unclear and adding explanation/data for those parts can improve the paper quality a lot. Please find them in the "detailed comments" section.

**Deanonymize Review:**

no

**Detailed Comments:**

Important:
- Publishing the implemented code for the proposed method would be very beneficial to understand the paper better and also could be very helpful for the research community in the field.
- It is written in the paper that using the original constant M_opt in the RandAument framework is not suitable for H&E stained images and the authors proposed a new scheme to find that as suggested on page 5. However, there is no comparison between these two methods in the results section to confirm this quantitatively.
- As the AUC results (in tables 3, 5, 6) for the proposed method and the method by Tellez et all are very competitive and in some cases, the differences are bellow 1%, statistical tests can be performed to show that the differences in the results are significant. (Suggestion: as AUC is used for comparison DeLong et al. method can be used. One implemented example can be found in MedCalc: https://www.medcalc.org/manual/comparison_of_roc_curves.php)

Minor comments:
- To adapt the RandAgment for H&E-stained images, some augmentation (posterize, solarize and invert) were excluded in the framework and some (HED- HSV) were added. However, in the discussion section, it is described that the superior performance of the handcrafted augmentation technique (by Tellez et al) in one of the cross-validation folds may be related to the additional augmentation steps such as Gaussian noise or elastic deformation. Why these two augmentations were not added to the adapted RandAugment framework to see if that can boost the classification performance?
- There is no reference for figure 2 in the text.
- In figure 1, the emphasize is on the variant image appearance based on the utilised staining techniques and scanners. It can be written in the image caption that the images belong to the same organ (and probably to the same disease type?) to show that the differences in the images are not coming from the differences in organs or disease type.




**Final Rating Justification:**

All my comments have been addressed adequately. I would like to thank the authors for revising the manuscript.

**Justification Of The Preliminary Rating:**

The paper presents a new approach for automatic data augmentation for H&E-stained images. The paper is well-written and the topic is very interesting for the research community. However, there are some unclear parts in the paper that can be modified to improve the paper quality.

**Paper Type:**

both

**Questions To Address In The Rebuttal:**

Please refer to the "weakness" and "detailed comment" sections.

**Special Issue:**

no

---

> ### Author Response · Authors · 2021-03-18
> **Reply to AnonReviewer3**
>
> We thank the reviewer for his or her feedback and time.  We will address the points  raised in the review, below.
>
> “Publishing the implemented code for the proposed method would be very beneficial to understand the paper better and also could be very helpful for the research community in the field.”
>
> Code will be available online at https://github.com/DIAGNijmegen/pathology-he-auto-augment. We need a couple of days to clean it up. The link is also included in the updated version of a manuscript.
>
> “It is written in the paper that using the original constant M_opt in the RandAument framework is not suitable for H&E stained images and the authors proposed a new scheme to find that as suggested on page 5. However, there is no comparison between these two methods in the results section to confirm this quantitatively”.
>
> We additionally include results of a model trained with a constant value of m as proposed in RandAugment . The model achieves an average AUC of 0.942 ± 0.028 on test sets, in comparison the model trained with our proposed augmentation method achieves an average AUC of 0.964± 0.014.  Please refer to the Appendix A in the updated version of a manuscript for the detailed results of the experiment.
>
> “As the AUC results (in tables 3, 5, 6) for the proposed method and the method by Tellez et all are very competitive and in some cases, the differences are bellow 1%, statistical tests can be performed to show that the differences in the results are significant. (Suggestion: as AUC is used for comparison DeLong et al. method can be used. One implemented example can be found in MedCalc: https://www.medcalc.org/manual/comparison_of_roc_curves.php).”
>
> Bootstrapping was used to assess the significance of the improvements introduced by the proposed method. The proposed method significantly (p<  0.05) outperforms the baseline (Tellez et all, 2019). We used the following package to compute p-value https://mateuszbuda.github.io/2019/04/30/stat.html.
>
> “To adapt the RandAgment for H&E-stained images, some augmentation (posterize, solarize and invert) were excluded in the framework and some (HED- HSV) were added. However, in the discussion section, it is described that the superior performance of the handcrafted augmentation technique (by Tellez et al) in one of the cross-validation folds may be related to the additional augmentation steps such as Gaussian noise or elastic deformation. Why these two augmentations were not added to the adapted RandAugment framework to see if that can boost the classification performance?”
>
> We noticed that simply increasing further a number of transforms did not yield expected improvements in performance, in some cases the performance was even lower. We assume that it might be related to the fact in our approach, as well as in Cubuk et al., 2020, the transforms are sampled with uniform probability 1/K, where K is a number of possible transforms in the list. On one hand, this assumption allows us to significantly reduce the search space. On the other hand, if certain augmentations introduce more benefit than others, simply adding elements to the list might reduce the impact of those “very positive” transforms. Thus, in future work we plan to increase the pool of transforms taking into account their relative importance.
>
>
> “There is no reference for figure 2 in the text.”
>
> We fixed it. Thank you.
>
> “In figure 1, the emphasize is on the variant image appearance based on the utilised staining techniques and scanners. It can be written in the image caption that the images belong to the same organ (and probably to the same disease type?) to show that the differences in the images are not coming from the differences in organs or disease type.”
>
> Fixed to: “Stain variation among different centers in tumour metastasis detection in breast lymph node tissue resections. Examples of WSI patches originating from different institutions (from left to write): cwh, lpe, rh, rumc, umcu.”
>
> David Tellez, Geert Litjens, P eter B ́andi, Wouter Bulten, John-Melle Bokhorst, Francesco Ciompi,and Jeroen van der Laak. Quantifying the effects of data augmentation and stain color normalization in convolutional neural networks for computational pathology. Medical Image Analysis, 58:101544, 2019. ISSN 1361-8415. doi: https://doi.org/10.1016/j.media.2019.101544

---

### Official Review · AnonReviewer4 · 2021-03-06

**Confidence:** 5
**Preliminary Rating:** 4
**Recommendation:** Poster
**Final Rating:** 4

**Summary:**

The paper proposes a modification of the RandAugmented framework that is tailored towards H&E-stained histopathology images. This paper addresses the problem of domain generalisation, which is very relevant in histopathology image analysis. An extensive set of experiments is performed, including a comparison to two baselines (no augmentation and previously reported best augmentation settings). Moderate improvement over the baseline is reported.

**Strengths:**

The proposed methodology is well motivated and a natural extension of the RandAugment framework for this type of image data. The problem that the method addresses is of primary importance in histopathology image analysis. The experimental setup is very extensive. The paper is very well written.

**Weaknesses:**

The improvements are somewhat moderate, however, I do not see this as a major weakness.  It would have been nice to also see a comparison to other methods that tackle domain generalisation in histopathology (such as stain normalisation).

**Deanonymize Review:**

no

**Detailed Comments:**

Minor improvement:
- Discussing Lafarge et al (2019), the authors state that "it is not possible to ensure robustness to variation which are not in the data used for domain adversarial training.". This method in fact does aim to learn domain invariant features that can potentially generalise to unseen domains.

**Final Rating Justification:**

I remain of the opinion that this is a strong paper and keep my original recommendation.

**Justification Of The Preliminary Rating:**

I do not have any major criticisms of the work. It is a well written paper that presents methodology for a very relevant problem. While the achieved results are moderate in comparison to the baseline, I find the work very valuable addition to the literature.

**Paper Type:**

methodological development

**Questions To Address In The Rebuttal:**

As mentioned before, it would be nice to see direct comparison to staining normalisation or a combination of data augmentation and stain normalisation.

**Special Issue:**

no

---

> ### Author Response · Authors · 2021-03-18
> **Reply to**
>
> AnonReviewer 4
>
> We appreciate the feedback and time spent on the review. We will address the points mentioned below.
>
> "Minor improvement:
> Discussing Lafarge et al (2019), the authors state that "it is not possible to ensure robustness to variation which are not in the data used for domain adversarial training.". This method in fact does aim to learn domain invariant features that can potentially generalise to unseen domains. "
>
> The authors agree with the statement that the method proposed by Lafarge et al. (2019) aims to learn domain invariant features that can potentially generalize to unseen domains. However, it is difficult to estimate the extent of such invariance as the learned representation depends on the limited number of variations (present in the training set) the network was exposed to during training.
>
> "Questions To Address In The Rebuttal:
> As mentioned before, it would be nice to see direct comparison to staining normalisation or a combination of data augmentation and stain normalisation. "
>
> We additionally include results of a model trained with no data augmentation but using only stain normalization. In this experiment we use the stain normalization method proposed by Tellez et al, (2019). The rest of the training parameters and setup remain unchanged. The model achieves an average AUC of 0.893 ± 0.067, in comparison the model trained with our proposed augmentation method achieves an average AUC of 0.964± 0.014. Please refer to Appendix B in the updated version of a manuscript for the detailed results of the experiment.
>
> [1]Maxime W. Lafarge, Josien P. W. Pluim, Koen A. J. Eppenhof, and Mitko Veta. Learning domain-invariant representations of histological images. Frontiers  in  Medicine , 6:162, 2019.  ISSN 2296-858X.  doi:  10.3389/fmed.2019.00162.
> [2]David Tellez, Geert Litjens, P eter B ́andi, Wouter Bulten, John-Melle Bokhorst, Francesco Ciompi,and Jeroen van der Laak. Quantifying the effects of data augmentation and stain color normalization in convolutional neural networks for computational pathology. Medical Image Analysis, 58:101544, 2019. ISSN 1361-8415. doi: https://doi.org/10.1016/j.media.2019.101544

---

### Official Review · AnonReviewer1 · 2021-03-08

**Confidence:** 5
**Preliminary Rating:** 3
**Recommendation:** Poster
**Final Rating:** 3

**Summary:**

This paper proposes an adaptation of RandAugment, an automatic data-augmentation method, to H&E histopathology data. With a predefined set of data augmentation methods, an automatic selection is performed to optimize the generalization of a small CNN to unseen centers (validation). Good results are obtained on the classification of cancerous patches in breast lymph nodes whole slide images.

**Strengths:**

The paper is globally well written, well structured and easy to follow (some typos and grammar mistakes to fix).
It proposes an important work for histopathology, dealing with a recurrent problem of domain adaptation.
The related work is good as well as the motivation for the method.
The results are good and compared to the SoA.


**Weaknesses:**

The technical novelty is limited as it is a rather simple adaptation of an existing method to the H&E data. However, it is sufficient in my view and the paper is relevant and useful to the MIDL community.


**Deanonymize Review:**

no

**Detailed Comments:**

I provide some comments and points to address in the following.

(1) In the introduction: “it is not possible to ensure robustness to variation which are not in the data used for domain-adversarial training.” It is true, but unlabelled test data can be added to the training, e.g. some slides from a new center can be added, for which only the domain adversarial loss is used. See e.g. Eq. (8) in [1] for general images and Eq. (1) in [2] for histopathology images.

(2) In the introduction, mention what are the specificities of H&E that motivate the adaptation of RandAugment.

(3) In 2.1:  “six-fold cross-validation” It is not really a six-fold cross-validation since the training is fixed and the validation and test are permuted. It could be confusing to describe it like this.

(4) In Figure 2: Can you not augment before cropping the patches? This would avoid the edge artifacts of “translate”, “shear” and “rotate” and would result in more meaningful transformations, less out-of-distribution.

(5) In Table 2: Is L2 for regularization? Why is it with all layers? Maybe remove it, mention it in the caption/text, and replace it by the feature map sizes.

(6) You could comment on the use of augmentation for dealing with imbalanced data, another common problem in histopathology.

[1] Ganin, Yaroslav, et al. "Domain-adversarial training of neural networks." The journal of machine learning research 17.1 (2016): 2096-2030.
[2] Graziani, Mara, et al. "Guiding CNNs towards Relevant Concepts by Multi-task and Adversarial Learning." arXiv preprint arXiv:2008.01478 (2020).


**Final Rating Justification:**

Thank you for the response. I maintain the weak accept rating.

**Justification Of The Preliminary Rating:**

It is not a groundbreaking paper, yet it is an important work that can be useful to many researchers working on H&E data often encountering generalization problems. It is well written, and the results are good. I suggest a poster presentation.

**Paper Type:**

methodological development

**Questions To Address In The Rebuttal:**

Please address the points mentioned above.

**Special Issue:**

no

---

> ### Author Response · Authors · 2021-03-18
> **Reply to AnonReviewer1:**
>
> We appreciate the reviewer for his or her time and feedback.  We will address the points raised in the review, below.
>
> “(1) In the introduction: “it is not possible to ensure robustness to variation which are not in the data used for domain-adversarial training.” It is true, but unlabelled test data can be added to the training, e.g. some slides from a new center can be added, for which only the domain adversarial loss is used. See e.g. Eq. (8) in [1] for general images and Eq. (1) in [2] for histopathology images.”
>
>
> We agree with a point mentioned by the reviewer. However, while adding unlabeled test data to the training can potentially improve robustness of the model to variations not present in the labeled training set, such approach would:
>
> 1) Require retraining of the model every time data from a new center is encountered.
> 2) Even though only the domain adversarial loss can be used, the weights of the model will be updated and it is not guaranteed that overall performance of the model will remain unchanged.
>
> “(2) In the introduction, mention what are the specificities of H&E that motivate the adaptation of RandAugment.”
>
> We added “We perform several modifications to RandAugment, enabling models to generalize over data originating from external institutions in computational pathology. In particular, we focus on increasing the robustness of a model to variations in colour properties of the image resulting from different H&E staining protocols and scanners” .
>
> “(3) In 2.1: “six-fold cross-validation” It is not really a six-fold cross-validation since the training is fixed and the validation and test are permuted. It could be confusing to describe it like this. “
>
> We agree with a point raised by the reviewer. We substitute the confusing “six-fold cross-validation” name  with description:
>
> “Thus, we arrange the experiments in the following way:  we always train the model only on data from rumc, the validation set contains a subset of rumc and data from two external institutions, while the test set consists of data from remaining two centers.  The datasets in validation and testing are subsequently permuted resulting in six possible unique combinations.”
>
> “(4) In Figure 2: Can you not augment before cropping the patches? This would avoid the edge artifacts of “translate”, “shear” and “rotate” and would result in more meaningful transformations, less out-of-distribution.”
>
> We believe that this is a very useful point. Unfortunately we haven't thought of it earlier ourselves. However it would be difficult to address in a timeframe of this rebuttal, as it would require redoing all experiments. We will definitely take it into account for future work.
>
> “(5) In Table 2: Is L2 for regularization? Why is it with all layers? Maybe remove it, mention it in the caption/text, and replace it by the feature map sizes.”
>
> We clarified the meaning of L2 in text:
>
> “We apply 1∗10^(−6) factor L2 regularization as in Tellez et al. (2019).”
>
> We decided to use a model and training parameters same as in the baseline to make comparison as fair as possible.
> We substituted the aforementioned column with output feature map sizes.
>
> “(6) You could comment on the use of augmentation for dealing with imbalanced data, another common problem in histopathology.”
>
> We added “Data  augmentation  is  a  frequently  used  solution  to  introduce  a  certain  degree  of  invariance to a CNN (Shorten and Khoshgoftaar, 2019), it can also be used to tackle class imbalance (Sebai et al., 2020).”
>
> [1]Connor Shorten and Taghi M. Khoshgoftaar. A survey on image data augmentation for deep learning.Journal of Big Data, 6(1):60, Jul 2019.  ISSN 2196-1115.  doi:  10.1186/s40537-019-0197-0.
> [2]Meriem Sebai, Xinggang Wang, and Tianjiang Wang.  Maskmitosis:  a deep learning framework forfully supervised, weakly supervised, and unsupervised mitosis detection in histopathology images.Medical & Biological Engineering & Computing, 58(7):1603–1623, Jul 2020. ISSN 1741-0444. doi:10.1007/s11517-020-02175-z

---

### Official Review · AnonReviewer2 · 2021-03-09

**Confidence:** 4
**Preliminary Rating:** 3
**Recommendation:** Poster
**Final Rating:** 3

**Summary:**

This paper proposes modification of RandAugment method for the augmentation of patch images extracted from digital pathology slide. By conducting experiment with multi-instutional study setting, the generalization performance using proposed augmentation strategy is evaluated. The performance was also compared with no-augmentation and manual augmentation setting which demonstrate the superiority of the proposed method.

**Strengths:**

- The paper tackles important and relevant task for digital pathology.
- The problem setting is reasonable for multi-institutional studies.
- The experimental result provided in Table 4-6 clearly demonstrate the benefit of the proposed method.

**Weaknesses:**

- Considering additional computational effort for the proposed augmentation strategy, the relative performance improvement seems marginal.
- Other automatic augmentation method should be also considered for performance benchmark.

**Deanonymize Review:**

no

**Detailed Comments:**

- In discussion, "rather then" -> "rather than"


**Final Rating Justification:**

Thank you for the response. I keep my rating as 'weak accept'


**Justification Of The Preliminary Rating:**

This paper proposes modification of RandAugment for improving the patch-wise classificaiton performance of digital pathology slide. Based on realistic setting, the paper evaluted the effectiveness of the proposed augmentation strategy over no-augmentation and manual augmentation strategy. There is some limitations such as the marginal performance improvement and lack of analysis in terms of computational cost.

**Paper Type:**

methodological development

**Questions To Address In The Rebuttal:**

- Please also describe additional computational cost incurred by the proposed automatic augmentation method.

**Special Issue:**

no

---

> ### Author Response · Authors · 2021-03-18
> **Reply to AnonReviewer2**
>
> We thank the reviewer for his or her feedback and time.  We will respond to the points that are raised in the review, below.
>
> “Considering additional computational effort for the proposed augmentation strategy, the relative performance improvement seems marginal.”
>
> This would be true if hyperparameter settings for data augmentation are ‘set-it-and-forget-it’, however, in general, most researchers spend a lot of time manually optimizing data augmentation strategies because of the performance increase such optimization typically results in.
>  As a baseline for comparison, we adopt the highest performing augmentation method from Tellez  et  al.,  (2019):  a  manually  tuned  combination of  classical  and  domain-specific data augmentation techniques which outperformed nearly 40 (mentioned in the Tellez  et  al.,  (2019) paper) combinations of various data augmentation/stain normalization strategies. Such extensive tuning takes at least as much of GPU time running experiments in addition to the human time investment related to manually optimizing parameters and still our method outperforms this manual strategy.
> Exact comparisons are sadly impossible because manual tuning time, such as in Tellez et al. (2019) is hardly ever reported.
>
> “Other automatic augmentation method should be also considered for performance benchmark.”
>
> In future we plan to compare against other automatic augmentation methods, however it is impossible to run the required experiments and perform a good analysis  in the short time frame of this rebuttal.
>
> “In discussion, "rather then" -> "rather than"”
> We fixed it. Thank you.
>
> “Please also describe additional computational costs incurred by the proposed automatic augmentation method.”
>
> The additional computational cost incurred by the proposed automatic augmentation method depends on the size of the explored search space and sampling strategy. For the model and dataset used in this study, the final  search of optimal m and n hyperparameter values took approximately 130 GPU hours. As stated before, exact comparisons to manual tuning are impossible, as this time is typically not reported by researchers, but we expect it to match or exceed the automated strategy in addition to including human engineering time.
>
> [1]David Tellez, Geert Litjens, P eter B ́andi, Wouter Bulten, John-Melle Bokhorst, Francesco Ciompi,and  Jeroen  van  der  Laak.   Quantifying  the  effects  of  data  augmentation  and  stain  color  normalization in convolutional neural networks for computational pathology. Medical  Image  Analysis, 58:101544, 2019.  ISSN 1361-8415.  doi:  https://doi.org/10.1016/j.media.2019.101544

---

### Meta-Review · Area_Chair1 · 2021-03-27

**Recommendation:** Accept (Poster)

**Metareview:**

The reviewers all acknowledge that the paper is well written, well structured and easy to follow, tackles important task for digital pathology,  is relevant and useful to the MIDL community and that the proposed method can be adapted for various image processing tasks in H&E-stained images.

Thus, even though technical novelty is limited and the performance is marginally improved, as also acknowledged, I recommend acceptance of this paper.

**Paper Type:**

validation/application paper

---

### Decision · Program_Chairs · 2021-03-31

Accept